# Nontraumatic Fat Embolism and Fat Embolism Syndrome in Patients with Purulent Bacterial Bronchopneumonia

**DOI:** 10.3390/jcm14176097

**Published:** 2025-08-28

**Authors:** Beáta Ágnes Borsay, Barbara Dóra Halasi, Róbert Kristóf Pórszász, Katalin Károlyi, Teodóra Tóth, Péter Attila Gergely

**Affiliations:** 1Institute of Forensic Medicine, Faculty of Medicine, University of Debrecen, Nagyerdei krt. 98, H-4032 Debrecen, Hungary; halasi.barbara@med.unideb.hu (B.D.H.); gergely.peter@med.unideb.hu (P.A.G.); 2Department of Pharmacology and Pharmacotherapy, Faculty of Medicine, University of Debrecen, Nagyerdei krt. 98, H-4032 Debrecen, Hungary; porszasz.robert@med.unideb.hu; 3Department of Pathology and Laboratory Medicine, Jefferson Einstein Hospital Philadelphia, 5501 Old York Rd, Philadelphia, PA 19141, USA; katalinkarolyi0@gmail.com; 4Department of Pathology, Kenézy Gyula Campus, Clinical Center, University of Debrecen, Bartók Béla út 2-26, H-4031 Debrecen, Hungary; teodoradrtoth@gmail.com

**Keywords:** fat embolism, fat embolism syndrome, elevated C-reactive protein level, fat globules forming in purulent bronchopneumonia, Oil Red O

## Abstract

**Background**: Fat embolism frequently occurs as a result of trauma, such as long bone fractures and orthopedic surgeries, as well as in certain non-traumatic conditions. The formation can be attributed to mechanical or biochemical processes. According to Hullman’s biochemical hypothesis, elevated C-reactive protein levels facilitate the precipitation of very-low-density lipoproteins and chylomicrons, forming fat globules that may result in fat embolism. Based on the abovementioned hypothesis, this study aims to detect fat embolism in autopsy patients (postmortem) suffering from bronchopneumonia and determine its possible role as a cause of death. **Methods**: A group of autopsies of deceased individuals with bacterial purulent bronchopneumonia with confirmed or presumed elevated C-reactive protein levels was rigorously selected, excluding those with other potential causes of fat embolism such as cardiopulmonary resuscitation, hypothermia, and diabetes mellitus. Multiple organs were sampled for frozen section analysis using Oil Red O fat staining and assessed for the presence and extent of fat embolism. The Falzi score, as modified by Janssen, was employed for the lung tissue. **Results**: In 73% of the cases, predominantly sporadic, Grade 0 or Grade I fat embolism was observed; however, in none of the cases was fat embolism identified as the cause of death or as a significant contributing factor. Furthermore, neither fat embolism syndrome nor multiorgan fat embolism were detected. **Conclusions**: Although an elevated C-reactive protein level facilitates the formation of fat globules and fat embolism, its role as a direct cause of mortality remains uncertain. It may predispose individuals to such conditions and potentially interact with other factors, such as minor soft tissue trauma, to exacerbate the severity of fat embolism or its clinical manifestations. These findings underscore the necessity for further comprehensive investigations within the contexts of infection/inflammation, fat embolism, and dyslipidemia.

## 1. Introduction

Fat embolism was initially identified in humans by Zenker in 1861, following a fatal thoracoabdominal injury sustained by a railroad worker [1]. It is a well-known phenomenon in forensic medicine practice following extensive soft tissue injuries (e.g., blunt trauma or burns) or long bone or pelvis fractures, and is also typical in polytraumatic cases. Orthopedic surgeries are also a risk factor for development. Some other conditions may also result in fat embolism, such as liposuction, parenteral lipid infusion, diabetes mellitus, osteomyelitis, alcoholic fatty liver disease, etc. (Table 1) [2,3,4,5,6,7,8,9,10,11,12,13].

Similarly to other types of embolisms, a fat embolism is considered a vital sign and is characterized by the presence of fat globules in the circulatory system. These fat globules may traverse the pulmonary circulation to reach systemic organs or pass from the right ventricle through the patent foramen ovale (PFO) [14]. Clinically, only a small fraction of fat embolism cases progress to fat embolism syndrome (FES), a potentially fatal condition. In the 1970s, the estimated mortality rate for FES was between 10% and 20%. However, due to improvements in early diagnostic methods and supportive care, this rate has now decreased to between 6% and 10% [15,16]. This phenomenon was first described in 1970 by Gurd [17]. The classical triad of FES manifests in 3–4% of cases, indicating that the mere presence of fat in the vessels does not necessarily result in significant clinical effects [18]. No single special diagnostic test is sensitive enough in clinical practice [16]. In certain instances, isolated pulmonary fat embolism may be observed, but the involvement of the central nervous system and skin is also a major criterion for FES, as defined by Gurd and Wilson (Table 2) [19,20,21,22].

Other scoring system examples include Sconfeld’s, Emson’s, and Lindeque’s [2,23]. Two primary theories have been proposed to explain the etiology of fat embolism syndrome: the mechanical theory and the biochemical theory. The mechanical theory posits that disruption of the bone marrow allows fat globules to enter torn venules, thereby eliciting an inflammatory response. Conversely, biochemical theory suggests that the degradation of embolized fat results in the production of fatty acids and other intermediaries with pro-inflammatory effects in the circulation, leading to symptom manifestation [24,25,26]. The release of toxic metabolites causes vascular endothelial injury, which increases the vessel wall permeability and reduces surfactant production. This process is associated with acute respiratory distress syndrome (ARDS) and significantly contributes to the formation of non-traumatic fat embolism [27,28]. The overall incidence of ARDS is about 2% [29]. Frozen sections and lipid staining techniques, such as Oil Red O or Sudan Black B, applied to a native tissue sample, are effective methods for demonstrating the presence of fat or the space left by its dissolved form in paraffin-embedded samples, using hematoxylin–eosin (H&E) staining (Figure 1).

In addition to the previously mentioned samples, Hulman observed that C-reactive protein (CRP) promotes the calcium-dependent agglutination of very-low-density lipoproteins (VLDL) and chylomicrons, potentially leading to the formation of fat macroglobules (5–20 μm in diameter) that are sufficiently large to induce non-traumatic fat embolism [30]. C-reactive protein is a non-specific inflammatory marker synthesized by the hepatocytes in the liver, initiated by the release of interleukin 1-β (IL1β) and interleukin-6 (IL6) of the activated macrophages and monocytes [31,32,33,34,35]. It is elevated in both acute and chronic inflammatory conditions, such as rheumatoid arthritis, bacterial infections, malignancies, and injuries, including in patients with (poly)trauma. This study aimed to determine the presence and quantify spontaneous fat embolisms in patients with bacterial bronchopneumonia and to ascertain their potential role as a cause of death.

While fat embolism is typically associated with traumatic injuries, this study reports on the incidental postmortem detection of low-grade fat embolism in a non-traumatic, inflammatory setting. Our findings raise awareness of its possible occurrence under unanticipated conditions, such as bacterial purulent bronchopneumonia, but do not provide evidence of clinical fat embolism syndrome or a direct pathophysiologic mechanism.

## 2. Materials and Methods

The study design was submitted to the Regional Institutional Research Ethics Committee, Clinical Center, University of Debrecen, for approval. Tissue samples were obtained from routine autopsies performed at the Department of Forensic Medicine, University of Debrecen, Hungary. Over the one-year period from 13 February 2024, to 13 February 2025, a total of 1188 autopsy cases (clinical or forensic) were evaluated on a rolling basis. Of these, 24 cases were identified as having bacterial purulent bronchopneumonia exclusively, without any other lesions leading to fat embolism (e.g., heat stroke, trauma, surgery, diabetes mellitus, bone tumor lysis, cardiopulmonary resuscitation, etc., as indicated by literature data), as seen in Table 1. The borderline or ambiguous cases were also excluded. Patients diagnosed with bacterial bronchopneumonia were presumed to have elevated C-reactive protein (CRP) levels or had confirmed high CRP levels. In the “presumed” cases, there was no evidence of coexisting immunomodulatory conditions (such as malignancy, HIV infection, AIDS, congenital immunodeficiency (e.g., thymic hypolasia, hyper IgM syndrome, selected IgA deficiency), or permanent immunomodulatory therapy), which can lead to the downregulation of the CRP forming generally, or in infectious or inflammatory cases. During the assessment, an additional nine cases were excluded for various reasons identified through a more thorough review of clinical findings, autopsy, and histological evaluation. Specimens were collected on a rolling basis for frozen evaluation and paraffin embedding 26 to 144 h post-mortem, during the autopsy, and prior to burial (Figure 2).

Two sets of tissue samples were collected, each measuring 1 cm × 1 cm × 1 cm. One set was fixed in formalin and embedded in paraffin for standard hematoxylin–eosin (H&E) staining, whereas the other set was prepared for frozen section analysis using Oil Red O staining. The tissue samples comprised five lung specimens (one from each lobe: two central, two peripheral, and one middle section), four brain specimens (right frontal lobe, corpus callosum, right nucleus caudatus, and left occipital lobe), a fragment of the left anterior heart wall, and two kidney specimens (one from each side).

Two-thirds of the specimens were stored in the freezer (−18 °C) for a maximum of 14 days, and one-third were processed immediately. After attaching the specimen to the chuck by cryomatrix (Epredia “cryomatrix” embedding resin, Fisher Scientific, Pittsburg, PA, USA, product code: 12542716), the tissue was merged in −70 °C petrol ether on dry ice to freeze it. The specimens were cut by cryotome (Thermo Scientific with cryogenic working place and cryogenic blade), and the frozen sections were transferred to smooth, room-temperature slides. This method creates electrostatic interactions, which help adhesion. After 1 day of drying, the lipid-staining procedure was started. The slides were rinsed with propylene glycol (Molar Chemicals, catalog number: 00060), and they were stained for 5 min with a propylene glycol solution of 0.7% Oil Red O (Sigma-Aldrich, Burlington, MA, USA, catalog number: O0625). This was followed by an 85% propylene glycol rinse and then a distilled water wash. Finally, the slides were stained with Mayer’s Alum Hematoxylin (Molar Chemicals, Halásztelek, Hungary, catalog number: 42514) for 2 min. Distilled water rinse was followed by running tap water bluing for 4 min, and the slides were covered with water-based mounting media (Mount Quick Aqueous, Bio-Optica, Milan, Italy, code: 05-1740). 

Histological examinations were conducted using a Leica light microscope (Wetzlar, Germany, DM2500) and Pannoramic 1000 Digital Scanner (3DHISTECH, Budapest, Hungary). Two forensic experts and pathologists independently analyzed the slides within 48 h. Fat embolism evaluation was performed at 100× magnification, utilizing scores developed by Falzi et al. and modified by Janssen, based on the extent and configuration of fat embolism (no fat embolism (0), mild (I), distinct (II), massive (III)) (Table 3) [36]. As a positive control, subepicardial fatty tissue of the heart or lung tissue from a polytraumatic deceased individual was used (Figure 3).

The reviewers were blinded to each other’s assessments but not to the available clinical information. As a common application of their routine work on the abovementioned pulmonary fat embolism score system, there was no mismatch between them, so consensus or third-party adjudication was not needed.

As a technical note, we tried to make frozen sections from formalin (VWR Chemicals, Radnor, PA, USA, Formaldehyde 4% aqueous solution, buffered, catalog number: 1.00496.9010)-fixed tissue samples (from 2 to 14 days fixation). However, the quality was much worse, especially in the case of brain tissue, compared with the method mentioned above.

We utilized artificial intelligence tools to improve English fluency and readability and to edit the graphical abstract.

## 3. Results

During the autopsy of the 15 patients (4 female, 11 male) with bacterial bronchopneumonia, the lungs were found to be fragile, dark red, heavy, and bulky, with occasional abscesses forming, in most cases with pus content in the lower and upper airways, and in some cases with subpleural petechiae. The stages of pneumonia were not the same; these varied from area to area, not like in the case of lobar pneumonia. The macroscopic diagnosis was confirmed by the histological findings (with hematoxylin–eosin staining and Gram, EPAS or Grocott methenamine silver staining if necessary); massive intra-alveolar exudate was observed with neutrophil leukocytes, erythrocytes, fibrin meshwork, the presence of bacteria or, in later stages, with macrophages (Figure 4). In seven cases, abscess formation in the lungs was proven.

In all instances, the cause of death was attributed to purulent bacterial bronchopneumonia resulting from septicemia, right heart failure, or respiratory failure. In five cases, C-reactive protein (CRP) levels were documented, ranging from 31.54 to 156.36 mg/L. Despite the limited sample size, neither a CRP level of 31.54 mg/L nor 48.48 mg/L was found to be correlated with fat embolism. Clinically, in laboratory analyses, and during autopsies, none of the cases exhibited signs of fat embolism syndrome (FES), such as confusion, petechial rash, or an elevated erythrocyte sedimentation rate. In 9 out of 15 autopsies (60%), sporadic, punctiform (Grade 0) fat embolisms were observed, and in 2 cases (13%), Grade I, teardrop-like, more dispersed fat embolisms of the lung were identified through microscopic examination following lipid staining (Figure 5). Only one case demonstrated sporadic cardiac fat embolism in addition to pulmonary fat embolism. In four cases, no fat embolism was present. One case involved a patient with medically managed dyslipidemia. The average age and median age were 68.3 years and 69 years, respectively, and the age ranged from 48 to 92 years (Table 4).

As an artifact, in some cases, lipofuscin (reddish-brownish pigment granules) of the heart was determined as a sign of previous oxidative stress besides fatty degeneration of the heart [37]. In a small part of the lung tissue samples, foamy macrophages were detected with Oil Red O.

## 4. Discussion

The central research question of this study, in accordance with Hulman’s hypothesis, is to ascertain whether fat embolism can be identified in unexpected clinical scenarios, particularly in deceased patients who had bacterial purulent bronchopneumonia. Our findings indicate that the occurrence of fat embolism is incidental and does not directly lead to fatal outcomes, nor does it result in potentially fatal clinical fat embolism syndrome, or confirm a direct pathophysiological mechanism.

The premortem diagnosis of fat embolism and fat embolism syndrome is generally challenging. The latter is more common in males who have undergone traumatic events and are between the ages of 10 and 40 years, such as in cases of long bone fractures (diaphysis of the femur) or multiplex fractures, where the formation of emboli due to traumatic background is more obvious [38]. We focused on the biochemical theory of fat embolism formation. Although sporadic fat embolisms were detected in the lungs, they were not considered relevant regarding the cause of death. In addition to the elevated CRP level, other factors may also facilitate the formation of a profuse amount of fat macroglobules. The untreated, accompanying dyslipidemia seems to be a good condition for further research, besides microtraumatic, non-excessive soft tissue injuries. Also, it would be practical to examine samples over a certain “cut-off” level of CRP and/or expand the research to a greater case number. The CRP is a non-specific, clinically widely used, pentameric, acute-phase protein. Although many of the patients suffered from other conditions (e.g., cardiovascular disease, heart failure, inflammatory diseases in other organs), which can contribute to the elevated CRP levels, the degree of increase is less pronounced than in the case of bacterial purulent bronchopneumonia. The acute bacterial infections cause markedly elevated CRP levels, besides major traumas.

In addition to the administration of antibiotics, the anergy state may also be a limitation of this study. Owing to terminal immunosuppression, the systemic inflammatory response may have been attenuated, potentially accounting for the absence of a correlation between CRP levels and fat embolism findings in our research. Still, postmortem CRP-level establishment may be a useful method to exclude the improper cases if it is not known from premortem data; however, we have to consider including the decreased CRP cases [39]. The postmortem level of CRP is lower than the antemortem ones, and they correlate well. Hemolysis, hemoconcentration, and postmortem blood clots limit the measurement, but the persistent presence (31 days) of the protein is favorable. The results of samples from peripheral (iliac vein, subclavian vein) or right or left heart chamber blood were concordant [40,41]. On the other hand, in certain healthy adults, increased CRP levels (at 10 mg/L) were observed [42,43]. In none of the cases was fat embolism syndrome or acute respiratory distress syndrome (ARDS) noted. The former is a well-known, generalized consequence of fat embolism, as a multiorgan syndrome. At the same time, the latter is predisposed by fat embolism via the free fatty acids as a fundamental factor of acute inflammatory reaction of lung parenchyma. Still, fat embolism is not the primary cause of death [44,45]. The role of different kinds of pathogens is also an interesting question, and may feature in the quantity of fat embolism. 

The prevention of pulmonary fat embolism with bacterial bronchopneumonia is challenging and the focus has to be on managing the predisposing factors and mitigating the systemic response. Appropriate respiratory support with early and adequate antibiotic therapy, besides balancing between hypovolemia with impaired perfusion and hypervolemia leading to increased pulmonary edema keeps the focus on a holistic approach in managing strategies to address both infection and systemic consequences.

Advanced imaging techniques, such as computed tomography (CT) or magnetic resonance imaging (MRI) with higher resolution and increased sensitivity, will be able to highlight the presence of a fat embolism before the appearance of systemic symptoms. Circulating fat globules or the accumulation of fat in target organs with subtle changes in CT, MRI or on X-rays are ideal training environments for artificial intelligence (AI) and machine learning for early detection and differentiation between signs of pneumonia or other complications.

The possibility of extending the sample collection period was evaluated; however, practical and institutional limitations required the study’s completion within a set timeframe. Expanding autopsy sampling was beyond the project’s scope and available resources. Our aim was to perform a targeted, time-constrained observational analysis. The dataset illustrates the real-world challenges related to the accessibility and practicality of obtaining postmortem samples. Age was not considered a criterion for the inclusion or exclusion of patients. All patients who met the preset criteria were included in the study. For future studies on this matter, a prospective study can be implemented with a focus on different cohorts like wider age groups, women, and ethnicities other than Caucasian. While a more diverse demographic representation would be preferable, this study offers valuable insights within its established boundaries. 

The samples represent the natural demographic conditions in terms of age distribution. The study population predominantly comprises elderly individuals, so the presence of bacterial purulent bronchopneumonia reflects the epidemiology of the disease in lethal cases. To improve the applicability of research findings, future studies should focus on exploring fat embolism in a variety of clinical scenarios, including those that are non-lethal.

One of our aims was to enhance scientific awareness regarding this understudied issue, rather than to undertake a comprehensive population-level analysis. In clinical medicine, if bacterial bronchopneumonia accompanies certain mentioned conditions (see Table 1) and fat embolism is observed, we must raise the possibility of a combined origin.

The interpretation of fat embolism findings—particularly in non-traumatic or biochemically mediated cases—requires closer integration between clinical and forensic domains. Clinical data regarding inflammatory markers, lipid profiles, and therapeutic interventions (e.g., corticosteroids, antibiotics, and fluid therapy) are often unavailable in the forensic setting, limiting the interpretation of postmortem observations. Conversely, forensic autopsy findings may offer retrospective diagnostic insights in clinically ambiguous cases of acute respiratory failure. Developing shared diagnostic criteria and facilitating data exchange between hospital departments and forensic institutes would strengthen the identification and understanding of fat embolism or fat embolism syndrome. Collaborative, interdisciplinary protocols could help unify premortem and postmortem findings, enabling the more accurate attribution of the cause of death in complex cases.

### Limitations of the Study

The study is subject to certain unintended limitations, including constraints on quality, an unequal gender ratio.

## Figures and Tables

**Figure 1 jcm-14-06097-f001:**
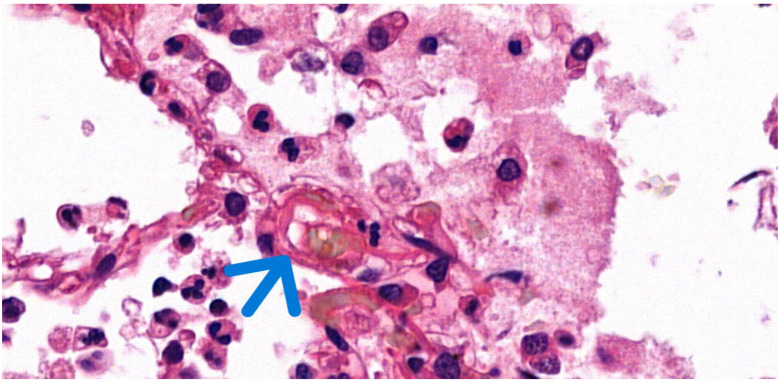
Histological image of pulmonary fat embolism with H&E staining. The visibly empty vacuoles (arrow) indicate the original location of the dissolved fat globules (63× magnification).

**Figure 2 jcm-14-06097-f002:**
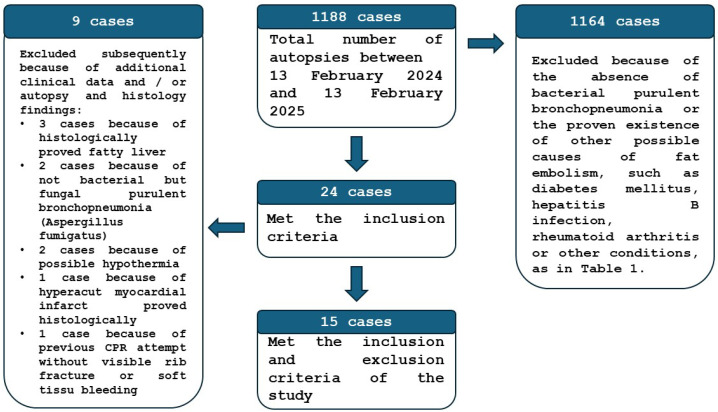
The flow diagram of the patient selection method of the study.

**Figure 3 jcm-14-06097-f003:**
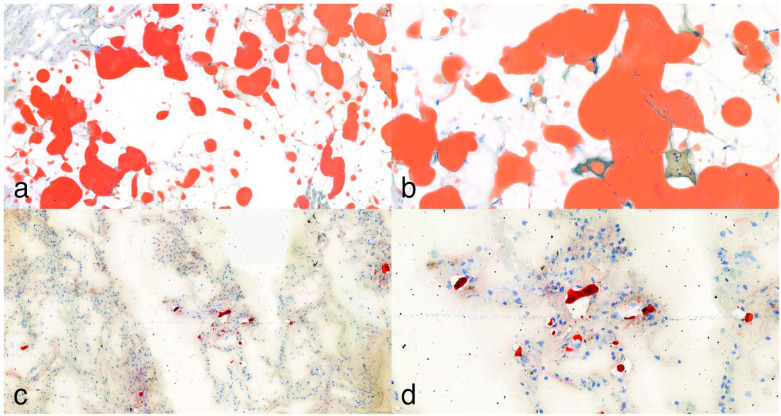
Oil Red O lipid staining. (**a**,**b**): In some cases, where applicable, staining of subepicardial fatty tissue served as a positive control, showing strong expression at 10× and 25× magnification. (**c**,**d**): Lipocytes present in frozen lung tissue sections of a polytrauma (pedestrian) victim at 10× and 25× magnification.

**Figure 4 jcm-14-06097-f004:**
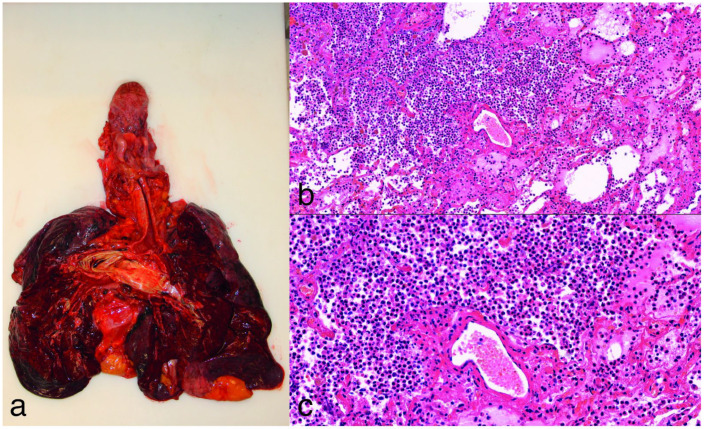
(**a**): Gross image of purulent bronchopneumonia. The lungs appear heavy, fragile, dark, and bulky. (**b**,**c**): Hematoxylin and eosin staining at 10× and 20× magnification of lung tissue, showing interalveolar fibrinopurulent exudate with neutrophils and intermixed macrophages. There is marked congestion, and the alveolar septa are widened with some reactive pneumocytes.

**Figure 5 jcm-14-06097-f005:**
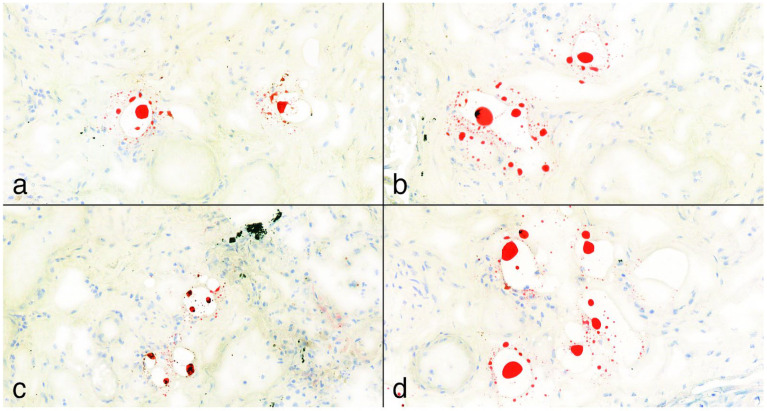
(**a**–**d**): Positive Oil Red O lipid staining samples from lungs at 25× magnification. The fat deposits appear mainly as punctiform structures with an isolated, patchy pattern and are not visible in every field of view. According to the pulmonary fat embolism scoring system by Falzi et al., modified by Janssen, the grading of these cases was 0. In image (**c**), a black artifact is visible, representing a carbon deposit.

**Table 1 jcm-14-06097-t001:** Possible causes of fat embolism.

Long Bones and the Pelvis Fracture	Parenteral Lipid Injection/Infusion
Orthopedic surgeries	Sickle cell disease
Soft tissue injuries	Prolonged corticosteroid therapy
Burn injuries	Hemorrhagic pancreatitis
Liposuction	Carbon tetrachloride poisoning
Heat stroke	Heart surgery
Hypothermia	Osteomyelitis
Diabetes mellitus	Fatty liver
Septicemia	Acute respiratory distress syndrome (ARDS)
Cardiopulmonary resuscitation (CPR)	Bone marrow necrosis
Bone tumor lysis	Thalassemia
Lung transplantation	Bone marrow transplant

**Table 2 jcm-14-06097-t002:** Major and minor criteria of fat embolism syndrome (FES), according to Gurd and Wilson, used as a clinical scoring system.

Major	Minor
Respiratory insufficiency-Hypoxia (<60 Hgmm O_2_) with bilateral radiographic changes	Tachycardia (>110 beats/minute)
Petechial rush	Pyrexia (>38.5 °C)
Cerebral involvement (confusion, coma, drowsiness)	Renal changes (fat globules in urine/lipiduria/anuria/oliguria)
	Emboli are visible in the retina on fundoscopy
	Jaundice
	Fat in sputum
	Drop in platelet count (thrombocytopenia) or hematocrit
	Increased ESR (erythrocyte sedimentation rate) > 71 mm/h

At least 2 major or 1 major and 4 minor criteria required for the diagnosis of FES.

**Table 3 jcm-14-06097-t003:** Pulmonary fat embolism scoring system based on Falzi et al., modified by Janssen (100× magnification).

Grade	Characteristic Features and Extent of Fat Embolism
0	no fat embolism or punctiform, sporadic, where relevant
1	mild, teardrop-like, fat embolism, with 25× magnification, observed in every field.
2	multiple, distinct, sausage or tear-like fat embolism, disseminated in every field
3	massive, antler-like fat embolism in all vision, every microscopic field

**Table 4 jcm-14-06097-t004:** Summary of the deceased individuals included in the study, as well as gender, age, autopsy findings, relevant clinical data, results of lipid staining and grading, and the known CRP level within 24 h of death.

No.	Sex	Age (Years)	Autopsy Findings	Clinical Data/Medical Record/ICD-11 (International Classification of Diseases)/Smoking Habits	Result of Lipid Staining with Oil Red O/Affected Organ(s)/Falzi Modified by Janssen Grade of Pulmonary Fat Embolism	Premortem CRP Level Within 24 h of Death CRP Level (mg/L) Reference: <5.2
1.	male	79	Purulent bronchitis and bronchiolitis, purulent pneumonia with abscess formation, lacunar status, shock kidney, severe athero- and arteriosclerosis, cachexia, decubitus, PEG (percutaneous endoscopic gastrostomy) implantation	Essential hypertension, unspecified (BA00.Z), Chronic kidney disease, unspecified (GB61.Z), Certain specified alcohol-induced mental or behavioral disorders (6C40.7), Dementia, unknown or unspecified cause (6D8Z), Simple renal cyst (GB80.0) Retention of urine (MF50.3), Feeding difficulties, unspecified (MG43.3Z) No data about smoking habits	No presence of fat embolism Grade 0	CRP: 48.48 mg/L
2.	female	55	Purulent bronchopneumonia with abscess formation, purulent bronchitis, pleurisy, focal lung edema, decubitus, mild atherosclerosis, hyaline perisplenitis, chronic congestion of the liver (nutmeg liver), gallstone, ischaemic stroke	Down syndrome (LD40.0) Non smoker	No presence of fat embolism Grade 0	ND (no data)
3.	male	57	Purulent bronchopneumonia with abscess formation, lung edema, brain edema, former ischaemic stroke in the right hemisphere, severe general athero- and coronary sclerosis, dilatation of the right heart atrium and ventricle, right adrenal gland cortex adenoma, prostatic hyperplasia, signs of former rib fracture and slashes of the right forearm	Schizophrenia, episode unspecified (6A20.Z), Chronic bronchitis, unspecified (CA20.1Z), Ischaemic heart diseases, unspecified (BA6Z), Essential hypertension, unspecified (BA00.Z), Epilepsy or seizures, unspecified (8A6Z), Bacterial pneumonia (CA40.0) Previous smoker	Sporadic fat embolism in the lung (punctiform) Grade 0	ND (no data)
4.	male	68	Fibrinous pleurisy, unspecified chronic bronchitis, purulent bronchopneumonia with abscess formation, mild-moderate atherosclerosis, prostatic hyperplasia, nephrosclerosis, cyst of kidney, gallstone, mild hypertrophy of the heart, dilatation of the right heart atrium and ventricle, goiter	Chronic bronchitis, unspecified (CA20.1Z), Gastro-esophageal reflux disease, unspecified (DA22.Z), Emphysema, unspecified (CA21.Z), Cholelithiasis, unspecified (DC11.Z), Certain specified alcohol-induced mental or behavioral disorders (6C40.7), Mixed depressive and anxiety disorder (6A73), Duodenal ulcer, unspecified (DA63.Z) Active smoker	No presence of fat embolism Grade 0	ND (no data)
5.	female	92	COVID pneumonia, secondary purulent bronchopneumonia, severe atherosclerosis, mild coronary sclerosis, cyst of the kidney, nephrosclerosis	Essential hypertension, unspecified (BA00.Z), Harmful effects of or exposure to noxious substances, chiefly nonmedicinal as to source, not elsewhere classified (NE61), Secondary psychotic syndrome, with delusions (6E61.1), COVID-19, virus identified (RA01.0), Dementia due to cerebrovascular disease (6D81) Non smoker	Sporadic fat embolism in the lung (punctiform) Grade 0	CRP: 156.36 mg/L
6.	male	75	Purulent bronchopneumonia, severe atherosclerosis, former hemorrhagic stroke in left temporal lobe, mild lung edema, goiter, prostate hyperplasia, decubitus, right femoropopliteal stent implant	Bacterial pneumonia, unspecified (CA40.0Z), Sepsis without septic shock (1G40), Gastro-esophageal reflux disease, unspecified (DA22.Z) No data about smoking habits	Sporadic fat embolism in the lung (punctiform) Grade 0	CRP: 133.96 mg/L
7.	male	48	Purulent bronchopneumonia, multiple decubital ulcers of lower extremities, cerebral edema, moderate coronary sclerosis, dilatation of the right heart atrium and ventricle, acute gastritis, and acute cystitis	In anamnesis: Concussion, unspecified (NA07,0Z), Chronic rhinosinusitis, unspecified (CA0A.Z), Intestinal infections due to Clostridioides difficile (1A04), Intervertebral disk degeneration, unspecified (FA80.Z), Intentional self-harm by exposure to or harmful effects of antiepileptics or antiparkinsonism drugs (PC97)-multiple times, Post anoxic brain damage (8E44) No data about smoking habits	No presence of fat embolism Grade 0	CRP: 31.54 mg/L
8.	male	67	Purulent bronchopneumonia with abscess formation, purulent bronchitis, and bronchiolitis, sclerotic valve of the thoracic aorta, congestion of the spleen, nephrosclerosis, catheter of the urinary bladder, PEG (percutaneous endoscopic gastrostomy) implantation, benign prostatic hyperplasia, goiter	Bacterial pneumonia unspecified (CA40.0Z), Respiratory failure, unspecified (CB41.2Z), Dementia due to cerebrovascular disease (6D81), Dementia due to Alzheimer’s disease (6D80), Bronchitis, unspecified (CA20.Z) Previous smoker	Sporadic fat embolism in the lung (punctiform) Grade 0	ND (no data)
9.	male	69	Purulent bronchopneumonia with abscess formation, chronic bronchitis, emphysema (unspecified), pulmonary edema, atherosclerotic heart disease, former hemorrhagic stroke in basal ganglia and cerebellum, nephrosclerosis, chronic cystitis, goiter, PEG (percutaneous endoscopic gastrostomy) implantation	Stroke not known if ischaemic or hemorrhagic (8B20), Pertrochanteric fracture of femur (left) (C72.31) in anamnesis more than half a year before death, Dementia due to cerebrovascular disease (6D81) Previous smoker	Sporadic fat embolism in the lung (punctiform) Grade 0	ND (no data)
10.	male	85	Purulent bronchopneumonia, chronic bronchitis with acute exacerbation, lung emphysema severe atherosclerosis and coronary sclerosis, atherosclerotic heart disease, lacunar infarction of the brain, liver congestion (nutmeg liver), congestion of the spleen, erosion of the gastric mucosa, diverticulosis of sigma bowel and colostomy forming, chronic cystitis with acute exacerbation, nephrosclerosis, prostatic hyperplasia and former TURP (transurethral prostate resection) operation, acute pyelonephritis, decubitus	Essential hypertension, unspecified (BA00.Z), Problems of the prostate (MF40.1), Hyperplasia of prostate (GA90), Surgical or postsurgical states, unspecified (QB6Z), Intervertebral disk degeneration of lumbar spine with prolapsed disk (FA80.9) No data about smoking habits	Sporadic fat embolism in the lung (punctiform) Grade 0	ND (no data)
11.	male	50	Early phase bronchopneumonia, edema of the lungs and acute congestion, dilatation of the heart right ventricle, alcoholic cardiomyopathy, shock kidneys	Mental and behavioral disorders due to use of alcohol: dependence syndrome (F10.2) Active smoker	Sporadic fat embolism in the lung (punctiform) Grade 0	ND (no data)
12.	female	75	Purulent bronchopneumonia, acute bronchitis, moderate atherosclerosis, mild atherosclerosis of the brain, brain edema, goiter, degeneration of the myocardium, PEG (percutaneous endoscopic gastrostomy) implantation cyst of the left kidney, chronic congestion of the liver (nutmeg liver), sacral decubitus	Feeding difficulties, unspecified (MG43.3Z), Bacterial pneumonia, unspecified (CA40.0Z), Essential hypertension, unspecified (BA00.Z), Dementia, unknown or unspecified cause (6D8Z), In anamneses: COVID-19, virus identified (RA01.0) No data about smoking habits	Punctiform and teardrop-like fat embolism of the lung, at 25-fold magnification, every field of vision, and sporadic fat embolism of the heart Grade I	ND (no data)
13.	female	69	Purulent bronchopneumonia with abscess formation, purulent bronchitis and bronchiolitis, subpleural petechiae, mild lung edema, goiter, decubitus in sacral region, mild general atherosclerosis	Dementia unknown or unspecified cause (6D8Z), Essential hypertension, unspecified (BA00.Z) No data about smoking habits	Sporadic fat embolism in the lung (punctiform) Grade 0	ND (no data)
14.	male	72	Purulent bronchopneumonia, severe atherosclerosis, prostatic hyperplasia, nephrosclerosis, lacunar status	Essential hypertension, unspecified (BA00.Z), Diabetes mellitus, type unspecified (5A14), Gastro-esophageal reflux disease, unspecified (DA22.Z) Carpal tunnel syndrome (8C10.0), Diaphragmatic hernia (DD50.0), Encephalopathy, not elsewhere classified (8E47), Hyperplasia of prostate (GA90) Non smoker	Mainly punctiform and in some views teardrop-like fat embolism of the lung, at 25-fold magnification, every field of vision, Grade I	ND (no data)
15.	male	64	Purulent bronchopneumonia, foreign material in the lower airways with abscess formation, acute bronchitis, cachexia, emphysema, chronic cystitis	Disorders of intellectual development, unspecified (6A00.Z), Dissociative disorders, unspecified (6B6Z), Acute bronchitis, unspecified (CA42.Z), Alcohol dependence, unspecified (6C40.2Z), Bacterial pneumonia, unspecified (CA40.0Z), Gastric ulcer, unspecified (DA60.Z), Cachexia, unspecified (MG20.Z) Active smoker	Sporadic fat embolism in the lung (punctiform) Grade 0	CRP: 53.84 mg/L

## Data Availability

The original contributions presented in this study are included in the article or at https://doi.org/10.5281/zenodo.15863297 (accessed on 13 July 2025). Further inquiries can be directed to the corresponding author.

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
