# Peer review of "Nontraumatic Fat Embolism and Fat Embolism Syndrome in Patients with Purulent Bacterial Bronchopneumonia"

_jcm, 2025, doi:10.3390/jcm14176097_

Round 1

Reviewer 1 Report

Comments and Suggestions for Authors

At line 42 the authors state "... Fat embolism was initially identified in humans by Zenker in 1981 ..." and at lines 53-55 they affirm "... . Clinically, only a small fraction of fat embolism cases progress to fat embolism syndrome (FES), a potentially fatal condition. In the 1970s, the estimated mortality rate for FES was between 10% and 20% ...". It is unclear to me how a fat syndrome could be described in 1970 and fat embolism described primarly in 1981.

At lines 97-99 the authors state "... The study focused on deceased patients diagnosed with bacterial bronchopneumonia, either primary or secondary, who were presumed to have elevated C-reactive protein (CRP) levels or had confirmed high CRP levels ...". In my opinion it is necessary to specify in which cases the elevated PRC is "presumed" and why: in case in which the data are missed I think it should be specified. Even because in the "discussion" section any reference to these cases in which the elevated CRP is elevated diappears at all. 

At line 100 the authors specify that they observed 24 cases: in my opinion it is necessary to specify how many autopsies reports they revisited to find the 24 cases and which was the method adopted to select them.

The authors state that "... Tissue samples were collected within a postmortem interval ranging from 26 to 144 hours ..." (line 112): as they affirm that 24 cases were selected and that the samples were collected whitin 26-144 hours it is unclear to me how the inclusion/exclusion could be made. In fact this description imply that in all the autopsy performed the authors used the described samples collection and after the histologic revision the included/excluded the case. On the contrary it should be argued that the study would be retrospective but in such case it would not be clear how the samples would be collected within 26-144 hours. In my opinion alle the section materials and methods needs of a deep revision and clarification.

The discussion section in my opinion does not discuss what the authors declare as the aims of the study. It is discussed if the relevation of high level of CRP in the post-mortem could be useful to further study and the possibility of study concerning more cases but it is real unclear which are the conclusions of the study and if the porposed hypothesis can be useful in the post-mortem diagnosis in the forensic field 

Author Response

For research article

Response to Reviewer 1 Comments

1. Summary

Thank you very much for taking the time to review this manuscript. Please find the detailed responses below and the corresponding revisions/corrections highlighted yellow in track changes in the re-submitted files.

2. Questions for General Evaluation

Reviewer’s Evaluation

Does the introduction provide sufficient background and include all relevant references?

Yes/Can be improved/Must be improved/Not applicable

Is the research design appropriate?

Yes/Can be improved/Must be improved/Not applicable

Are the methods adequately described?

Yes/Can be improved/Must be improved/Not applicable

Are the results clearly presented?

Yes/Can be improved/Must be improved/Not applicable

Are the conclusions supported by the results?

Yes/Can be improved/Must be improved/Not applicable

Are all figures and tables clear and Yes/Can be improved/Must

well presented? be improved/Not applicable

3. Point-by-point response to Comments and Suggestions for Authors

Comment 1: At line 42 the authors state "... Fat embolism was initially identified in humans by Zenker in 1981 ..." and at lines 53-55 they affirm "... . Clinically, only a small fraction of fat embolism cases progress to fat embolism syndrome (FES), a potentially fatal condition. In the 1970s, the estimated mortality rate for FES was between 10% and 20% ...". It is unclear to me how a fat syndrome could be described in 1970 and fat embolism described primarly in 1981.

Respond 1: Thank you for pointing this out; it was an erratum. We have corrected the 1981 to 1861 in line 47.

Comment 2: At lines 97-99 the authors state "... The study focused on deceased patients diagnosed with bacterial bronchopneumonia, either primary or secondary, who were presumed to have elevated C-reactive protein (CRP) levels or had confirmed high CRP levels ...". In my opinion it is necessary to specify in which cases the elevated PRC is "presumed" and why: in case in which the data are missed I think it should be specified. Even because in the "discussion" section any reference to these cases in which the elevated CRP is elevated diappears at all. 

Respond 2: In the „presumed” cases, there was no evidence of coexisting immunomodulatory conditions (such as malignancy, HIV infection, AIDS, congenital immunodeficiency (e.g, thymic hypolasia, hyper IgM syndrome, selected IgA deficiency) permanent immunomodulatory therapy) which can lead to the downregulation of the CRP forming generally, also in infectious or inflammatory cases, in lines 126-131.

There is also information by individually about the CRP status or levels in Table 4., the last column.

Comment 3: At line 100 the authors specify that they observed 24 cases: in my opinion it is necessary to specify how many autopsies reports they revisited to find the 24 cases and which was the method adopted to select them.

Comment 4: The authors state that "... Tissue samples were collected within a postmortem interval ranging from 26 to 144 hours ..." (line 112): as they affirm that 24 cases were selected and that the samples were collected whitin 26-144 hours it is unclear to me how the inclusion/exclusion could be made. In fact this description imply that in all the autopsy performed the authors used the described samples collection and after the histologic revision the included/excluded the case. On the contrary it should be argued that the study would be retrospective but in such case it would not be clear how the samples would be collected within 26-144 hours. In my opinion, alle the section materials and methods needs of a deep revision and clarification.

Respond 3, 4: The answer is the same for these questions. We have edited a flow-chart as a new Figure 2. (line 136), for a clearer understanding for patient selection. We have rephrased the Materials and Methods section 1st paragraph in order to clear this procedure, in lines 116-135.

Comment 5: The discussion section in my opinion does not discuss what the authors declare as the aims of the study. It is discussed if the relevation of high level of CRP in the post-mortem could be useful to further study and the possibility of study concerning more cases but it is real unclear which are the conclusions of the study and if the porposed hypothesis can be useful in the post-mortem diagnosis in the forensic field.

Respond 5: We appreciate the reviewer’s suggestion. We have completed the discussion section according to the reviwer’s proposal, in lines 309-319.

…The interpretation of fat embolism findings—particularly in non-traumatic or biochemically mediated cases—requires closer integration between clinical and forensic domains. Clinical data regarding inflammatory markers, lipid profiles, and therapeutic interventions (e.g., corticosteroids, antibiotics, and fluid therapy) are often unavailable in the forensic setting, limiting the interpretation of postmortem observations. Conversely, forensic autopsy findings may offer retrospective diagnostic insights in clinically ambiguous cases of acute respiratory failure. Developing shared diagnostic criteria and facilitating data exchange between hospital departments and forensic institutes would strengthen the identification and understanding of fat embolism or fat embolism syndrome. Collaborative, interdisciplinary protocols could help unify premortem and postmortem findings, enabling the more accurate attribution of the cause of death in complex cases…

4. Additional clarifications

  • We have ordered professional English editing by MDPI, based on which the English was improved..
  • There were no details about the „figures and tables” improvement, but we hope that the edited chart-flow (Figure 2.) added value for the study and upgraded the quality.

Reviewer 2 Report

Comments and Suggestions for Authors

The present work is an original and interesting article that seeks to establish the possible relationship between patients who died from bronchopneumonia, elevated C-reactive protein (CRP) and pulmonary embolism. The study is consistent, methodologically sound, and based on samples obtained from deceased subjects (post-mortem analysis).

This research is considered to contribute significantly to the understanding of the pathophysiology of embolism in contexts of septic systemic inflammation. Although the results are not conclusive, the methodology used is valid and rigorous and constitutes a relevant starting point for future studies that delve into this phenomenon.

It is suggested to review and strengthen the abstract, since it does not currently clearly communicate the objective of the study or the type of sample analyzed. It is important to note that tissues from deceased patients were used, which adds value to the pathological and biochemical analysis of the relationship between inflammation, dyslipidemia and thromboembolism.

Author Response

For research article

Response to Reviewer 2 Comments

1. Summary

Thank you very much for taking the time to review this manuscript. Please find the detailed responses below and the corresponding revisions/corrections highlighted yellow in track changes in the re-submitted files.

2. Questions for General Evaluation

Reviewer’s Evaluation

Does the introduction provide sufficient background and include all relevant references?

Yes/Can be improved/Must be improved/Not applicable

Is the research design appropriate?

Yes/Can be improved/Must be improved/Not applicable

Are the methods adequately described?

Yes/Can be improved/Must be improved/Not applicable

Are the results clearly presented?

Yes/Can be improved/Must be improved/Not applicable

Are the conclusions supported by the results?

Yes/Can be improved/Must be improved/Not applicable

Are all figures and tables clear and Yes/Can be improved/Must

well presented? be improved/Not applicable

Point-by-point response to Comments and Suggestions for Authors

Comment 1: The present work is an original and interesting article that seeks to establish the possible relationship between patients who died from bronchopneumonia, elevated C-reactive protein (CRP) and pulmonary embolism. The study is consistent, methodologically sound, and based on samples obtained from deceased subjects (post-mortem analysis).

This research is considered to contribute significantly to the understanding of the pathophysiology of embolism in contexts of septic systemic inflammation. Although the results are not conclusive, the methodology used is valid and rigorous and constitutes a relevant starting point for future studies that delve into this phenomenon.

It is suggested to review and strengthen the abstract, since it does not currently clearly communicate the objective of the study or the type of sample analyzed. It is important to note that tissues from deceased patients were used, which adds value to the pathological and biochemical analysis of the relationship between inflammation, dyslipidemia and thromboembolism.

Response 1: Thank you for pointing this out. We have reedited and competed the “Abstract” section according to reviwer’s proposal in lines 24-26, 40-42.

…Based on the abovementioned hypothesis, this study aims to detect fat embolism in autopsy patients (postmortem) suffering from bronchopneumonia and determine its possible role as a cause of death…

… These findings underscore the necessity for further comprehensive investigations within the contexts of infection/inflammation, fat embolism, and dyslipidemia….

Reviewer 3 Report

Comments and Suggestions for Authors

Well-written report on the occurrence of fat embolism and its possible relation to mortality in patients w bronchopneumonia.

Comments:  Detailed description of pathologic/histologic methods.

Specific comments:

  1. Table 4 -- Did any of the patients undergo cardiopulmonary resuscitation?  This has been frequently reported to lead to fat embolization and FE syndrome. Several patients were in their 40s and 50s -- one would imagine that attempts were made to resuscitate them.
  2.  Given that many patients had atherosclerosis, strokes and coronary artery disease, please indicate which of the patients in Table 4 smoked (current or in the past).
  3. Table 5, patient no. 5 -- which nonmedicinal drugs was she taking?
  4. Patient 7 -- said to have C. difficile infection.  Was she receiving prolonged antibiotics?
  5. P. 12 -- Regarding the CRP, while the authors allude to the causes of CRP elevation, they should emphasize that CRP is a nonspecific biomarker.  Many of their patients had underlying conditions other than pneumonia contributing to the large range of serum CRP levels -- heart failure, infections and inflammation in other organs.
  6. Was there documentation of serum triglyceride levels that might account for the FE?

Author Response

For research article

Response to Reviewer 3 Comments

1. Summary

Thank you very much for taking the time to review this manuscript. Please find the detailed responses below and the corresponding revisions/corrections highlighted yellow in track changes in the re-submitted files.

2. Questions for General Evaluation

Reviewer’s Evaluation

Does the introduction provide sufficient background and include all relevant references?

Yes/Can be improved/Must be improved/Not applicable

Is the research design appropriate?

Yes/Can be improved/Must be improved/Not applicable

Are the methods adequately described?

Yes/Can be improved/Must be improved/Not applicable

Are the results clearly presented?

Yes/Can be improved/Must be improved/Not applicable

Are the conclusions supported by the results?

Yes/Can be improved/Must be improved/Not applicable

Are all figures and tables clear and Yes/Can be improved/Must

well presented? be improved/Not applicable

Point-by-point response to Comments and Suggestions for Authors

Comment 1: Table 4 -- Did any of the patients undergo cardiopulmonary resuscitation?  This has been frequently reported to lead to fat embolization and FE syndrome. Several patients were in their 40s and 50s -- one would imagine that attempts were made to resuscitate them.

Respond 1: The CPR was an exclusion criterion. Please see the ”Materials and Methods” section 1st paragraph in line 123.

Comment 2: Given that many patients had atherosclerosis, strokes and coronary artery disease, please indicate which of the patients in Table 4 smoked (current or in the past).

Respond 2: We have added this information to Table 4.

Comment 3: Table 5, patient no. 5 -- which nonmedicinal drugs was she taking?

Respond 3: She was exposed to chlorine gas, resulting in poisoning.

Comment 4: Patient 7 -- said to have C. difficile infection. Was she receiving prolonged antibiotics?

Respond 4: Clinical data indicate that the patient was diagnosed with Clostridium difficile infection accompanied by pseudomembranous colitis more than one year prior to the time of death. The patient was successfully treated, and there was no extended administration of antibiotics, such as Vancomycin, around the time of death.

Comment 5: P. 12 -- Regarding the CRP, while the authors allude to the causes of CRP elevation, they should emphasize that CRP is a nonspecific biomarker.  Many of their patients had underlying conditions other than pneumonia contributing to the large range of serum CRP levels -- heart failure, infections and inflammation in other organs.

Respond 5: Thank you for this point. We have added this information to the manuscript.

The CRP is a non-specific, clinically widely used, pentameric, acute-phase protein. Although many of the patients suffered from other conditions (e.g., cardiovascular disease, heart failure, other organ inflammatory diseases), which can contribute to the elevated CRP levels, the degree of increase is less pronounced than in the case of bacterial purulent bronchopneumonia. The acute bacterial infections cause markedly elevated CRP levels besides major traumas. Please, see lines 251-256 in the manuscript.

Comment 6: Was there documentation of serum triglyceride levels that might account for the FE?

Respond 6: There was no documentation of serum triglyceride levels that might account for the fat embolism.

Reviewer 4 Report

Comments and Suggestions for Authors

This paper shows that unexpected fat embolism was observed in 15 cases among 19 patients who died of bacterial bronchopneumonia and underwent autopsy. One major limitation of this manuscript is the small number of cases. The authors are fully aware of this limitation, so I assume they did not perform statistical analysis in this manuscript. I believe this assumption is correct. However, I believe there are some significant issues that need to be addressed in this paper.

  1. The authors suggest that unexpected, low-grade fat embolisms may occur in non-traumatic, inflammatory conditions such as purulent bronchopneumonia. These embolisms may be mediated by inflammation-related mechanisms, such as elevated C-reactive protein (CRP). While this premise is plausible, the authors should make it clearer in the introduction and discussion that their main finding is the incidental detection of fat embolism in an unanticipated clinical setting rather than evidence of clinical fat embolism syndrome or a direct pathophysiological mechanism. Clarifying this point would improve the focus and scientific value of the study.

  1. The exclusion criteria and case selection process lack clarity and transparency. Although the authors state that nine cases were excluded due to confounding factors, such as cardiopulmonary resuscitation (CPR), hypothermia, or diabetes mellitus, the rationale for exclusion is only briefly described and seems subjective. Furthermore, it is unclear how many cases were excluded for each reason and whether borderline or ambiguous cases were discussed among the readers. To enhance reproducibility and transparency, the authors should provide a detailed breakdown of the exclusion reasons and the corresponding number of cases. 1) A detailed breakdown of the exclusion reasons and the corresponding number of cases excluded for each reason; and 2) A summary figure (flow diagram) showing the total number of autopsies assessed, the number of exclusions (with reasons), and the final number of included cases. Providing this information in either the main text or as a figure will significantly improve the study's methodological rigor and allow readers to critically assess the study population.

  1. Although the evaluation of fat embolism severity was performed by two independent observers, the manuscript does not address how interobserver variability was assessed or mitigated. To reduce the risk of observer bias, please clarify whether the reviewers were blinded to clinical information or to each other’s assessments. And also clarify how discrepancies were resolved (by consensus, third-party adjudication or otherwise). If such analyses were not performed, the authors should acknowledge this as a limitation and discuss its potential impact on grading reliability and data interpretation.

  1. The authors acknowledge the absence of younger age groups as a limitation of the study. However, given the context of autopsy cases of purulent bronchopneumonia, the fact that the study population predominantly comprises elderly individuals is both expected and appropriate. The absence of younger individuals does not constitute a methodological limitation, but rather reflects the epidemiology of the disease in fatal cases. I suggest revising this part of the 'Limitations' section to avoid framing the natural demographic distribution as a deficiency. Instead, the authors could emphasise the need to explore fat embolism in different clinical contexts or non-lethal cases in future studies if they wish to achieve broader generalisability.

Author Response

For research article

Response to Reviewer 4 Comments

1. Summary

Thank you very much for taking the time to review this manuscript. Please find the detailed responses below and the corresponding revisions/corrections highlighted yellow in track changes in the re-submitted files.

2. Questions for General Evaluation

Reviewer’s Evaluation

Does the introduction provide sufficient background and include all relevant references?

Yes/Can be improved/Must be improved/Not applicable

Is the research design appropriate?

Yes/Can be improved/Must be improved/Not applicable

Are the methods adequately described?

Yes/Can be improved/Must be improved/Not applicable

Are the results clearly presented?

Yes/Can be improved/Must be improved/Not applicable

Are the conclusions supported by the results?

Yes/Can be improved/Must be improved/Not applicable

Are all figures and tables clear and Yes/Can be improved/Must

well presented? be improved/Not applicable

3.Point-by-point response to Comments and Suggestions for Authors

Comment 1: The authors suggest that unexpected, low-grade fat embolisms may occur in non-traumatic, inflammatory conditions such as purulent bronchopneumonia. These embolisms may be mediated by inflammation-related mechanisms, such as elevated C-reactive protein (CRP). While this premise is plausible, the authors should make it clearer in the introduction and discussion that their main finding is the incidental detection of fat embolism in an unanticipated clinical setting rather than evidence of clinical fat embolism syndrome or a direct pathophysiological mechanism. Clarifying this point would improve the focus and scientific value of the study.

Respond 1: We appreciate the reviewer’s suggestion, according to it we have completed the „Introduction” and „Discussion” section. Please see lines 109-113 and 234-239 in the manuscript.

Comment 2: The exclusion criteria and case selection process lack clarity and transparency. Although the authors state that nine cases were excluded due to confounding factors, such as cardiopulmonary resuscitation (CPR), hypothermia, or diabetes mellitus, the rationale for exclusion is only briefly described and seems subjective. Furthermore, it is unclear how many cases were excluded for each reason and whether borderline or ambiguous cases were discussed among the readers. To enhance reproducibility and transparency, the authors should provide a detailed breakdown of the exclusion reasons and the corresponding number of cases. 1) A detailed breakdown of the exclusion reasons and the corresponding number of cases excluded for each reason; and 2) A summary figure (flow diagram) showing the total number of autopsies assessed, the number of exclusions (with reasons), and the final number of included cases. Providing this information in either the main text or as a figure will significantly improve the study's methodological rigor and allow readers to critically assess the study population.

Respond 2: The exclusion criteria are not subjective, they are based on literature data, references [2-13], in lines 121-124. The Table 1. contains the other potential causes of fat embolism. The borderline or ambiguous cases were excluded, please see in lines 124-125. We agree with this comment, that a flow diagram can improve the study’s quality and transparency. So, according to the reviewer’s suggestion, we have edited a flow chart and adopted it in the manuscript as Figure 2.

Comment 3: Although the evaluation of fat embolism severity was performed by two independent observers, the manuscript does not address how interobserver variability was assessed or mitigated. To reduce the risk of observer bias, please clarify whether the reviewers were blinded to clinical information or to each other’s assessments. And also clarify how discrepancies were resolved (by consensus, third-party adjudication or otherwise). If such analyses were not performed, the authors should acknowledge this as a limitation and discuss its potential impact on grading reliability and data interpretation.

Respond 3: We have completed thi information.

…The reviewers were blinded to each other's assessments but were not to the available clinical information. As a common application of their routine work of the abovementioned pulmonary fat embolism score system, there was no mismatch between them, so consensus or third-party adjudication was not needed…

Please see in lines 176-179.

Comment 4: The authors acknowledge the absence of younger age groups as a limitation of the study. However, given the context of autopsy cases of purulent bronchopneumonia, the fact that the study population predominantly comprises elderly individuals is both expected and appropriate. The absence of younger individuals does not constitute a methodological limitation, but rather reflects the epidemiology of the disease in fatal cases. I suggest revising this part of the 'Limitations' section to avoid framing the natural demographic distribution as a deficiency. Instead, the authors could emphasise the need to explore fat embolism in different clinical contexts or non-lethal cases in future studies if they wish to achieve broader generalisability.

Respond 4: We have reedited the „Discussion” and „Limitations of the study” sections. Plese, see lines 299-304, and 322-323.

…The samples represent the natural demographic conditions in terms of age distribution. The study population predominantly comprises elderly individuals, so the presence of bacterial purulent bronchopneumonia reflects the epidemiology of the disease in lethal cases. To improve the applicability of research findings, future studies should focus on exploring fat embolism in a variety of clinical scenarios, including those that are non-lethal…

…The study is subject to certain unintended limitations, including constraints on quality, an unequal gender ratio…

4. Additional clarifications

  • We have ordered professional English editing by MDPI, based on which the English was improved..
  • There were no details about the „figures and tables” improvement, but we hope that the edited chart-flow (Figure 2.) added value for the study and upgraded the quality.
  • .

Round 2

Reviewer 4 Report

Comments and Suggestions for Authors

I believe that the authors responded appropriately to my previous comments.